# Understanding family-level decision-making when seeking access to acute surgical care for children: Protocol for a cross-sectional mixed methods study

Bria Hall[1], Allison Tegge[1], Cesia Cotache Condor[2,3,4], Marie Rhoads[1], Terri-Ann Wattsman[5], Angelica Witcher[1], Elizabeth Creamer[6], Anna Tupetz[2,7], Emily R. Smith[2,3,4,7], Mamata Reddy Tokala[9], Brian Meier[1,8], Henry E. Rice[2,3,4]*

1 Virginia Tech Carilion School of Medicine, Roanoke, VA, United States of America, 2 Duke Global Health Institute, Durham, NC, United States of America, 3 Department of Surgery, Duke University School of Medicine, Durham, NC, United States of America, 4 Duke Center for Global Surgery and Health Equity, Duke University School of Medicine, Durham, NC, United States of America, 5 Department of Surgery, Carilion Clinic, Roanoke, VA, United States of America, 6 School of Education, Virginia Tech University, Blacksburg, VA, United States of America, 7 Department of Emergency Medicine, Duke University School of Medicine, Durham, NC, United States of America, 8 Department of Emergency Medicine, Carilion Clinic, Roanoke, VA, United States of America, 9 Health Analytics Research Team, Carilion Clinic, Roanoke, VA, United States of America

* henry.rice@duke.edu

**Data Availability Statement:** Data sharing is not applicable to this paper as no datasets were generated or analyzed. All relevant data from this

# Abstract

## Background

There is limited understanding of how social determinants of health (SDOH) impact family decision-making when seeking surgical care for children. Our objectives of this study are to identify key family experiences that contribute to decision-making when accessing surgical care for children, to confirm if family experiences impact delays in care, and to describe differences in family experiences across populations (race, ethnicity, socioeconomic status, rurality).

## Methods

We will use a prospective, cross-sectional, mixed methods design to examine family experiences during access to care for children with appendicitis. Participants will include 242 parents of consecutive children (0–17 years) with acute appendicitis over a 15-month period at two academic health systems in North Carolina and Virginia. We will collect demographic and clinical data. Parents will be administered the Adult Responses to Children's Symptoms survey (ARCS), the child and parental forms of the Adverse Childhood Experiences (ACE) survey, the Accountable Health Communities Health-Related Social Needs Screening Tool, and Single Item Literacy Screener. Parallel ARCS data will be collected from child participants (8–17 years). We will use nested concurrent, purposive sampling to select a subset of families for semi-structured interviews. Qualitative data will be analyzed using thematic analysis and integrated with quantitative data to identify emerging themes that inform a

study, that is not ethically restricted, will be made available upon study completion.

**Funding:** BH and MR received the standard graduate student funding from Virginia Tech Carilion School of Medicine to support subject compensation for this study (see funding letter for reference). The funders had no role in study design, data collection and analysis, decision to publish, or preparation of the manuscript.

**Competing interests:** The authors have declared that no competing interests exist.

**Abbreviations:** ARCS, Adult Responses to Children's Symptoms; ACE, Adverse Childhood Experiences; CONSORT, Consolidated Standards of Reporting Trials; COREQ, Consolidated Criteria for Reporting Qualitative Research; EMR, Electronic Medical Record; QUAL, Qualitative; QUANT, Quantitative; REDCap, Research Electronic Data Capture; SES, Socioeconomic status; SDM, Shared Decision-Making; SDOH, Social Determinants of Health; SILS, Single Item Literacy Screener; SSI, Semi-structured Interview; STROBE, Strengthening the Reporting of Observational Studies in Epidemiology.

conceptual model of family-level decision-making during access to surgical care. Multivariate linear regression will be used to determine association between the appendicitis perforation rate and ARCS responses (primary outcome). Secondary outcomes include comparison of health literacy, ACEs, and SDOH, clinical outcomes, and family experiences across populations.

## Discussion

We expect to identify key family experiences when accessing care for appendicitis which may impact outcomes and differ across populations. Increased understanding of how SDOH and family experiences influence family decision-making may inform novel strategies to mitigate surgical disparities in children.

## Introduction

### Background

Disparities in surgical access, delivery, and outcomes differ across many populations in the U. S., including gender, race, socioeconomic status (SES), health literacy, and residence in rural versus urban settings [1, 2]. In childhood conditions such as obesity, asthma, and cancer, parental experiences such as response to childhood symptoms, adverse childhood experiences (ACEs), health literacy, and various social determinants of health (SDOH) are associated with delays in care [3, 4]. Despite an increasing recognition of the impact of these experiences on clinical outcomes, there is a limited understanding of how these factors impact families' decisions to seek care. Most studies have focused on demographic descriptions using population-level data, while the valuable experiences of families themselves remain poorly understood.

Appendicitis is an ideal proxy condition to examine disparities for surgical care of children [5]. Appendicitis is one of the most common pediatric surgical emergencies, with an estimated 70,000 appendectomies performed annually in children in the U.S. [6].Using population-based data, children from low-income households, rural areas, or of ethnic and racial minorities are at increased risk for delays in care with higher rates of perforation, morbidity, and medical costs [2, 7–10].

Factors contributing to surgical disparities occur across multiple levels, including processes at the system, provider, and family/patient levels [11]. These multiple factors interact across the entire continuum of care, although family-level factors predominate at the point of seeking access to care [12]. While there are efforts to improve health disparities through equitable shared decision-making (SDM) between families and providers, at the point of seeking access to acute surgical care, children and their parents are often autonomous in their decision-making [13]. The drivers of parental decisions about when, where, and how to seek surgical care for children remain poorly defined. Decision-making while seeking surgical care is particularly complicated in young or non-verbal children who may struggle to communicate their symptoms, and children's perspectives on these processes are poorly described [14]. Understanding the experiences of parents and children using the valuable voices of families themselves and testing preliminary hypotheses for associations may facilitate intervention development that can help support high-quality, equitable surgical care for children.

## Objectives

This study will focus on defining family experiences during access to surgical care using the proxy condition of appendicitis at two regional academic children's centers in North Carolina and Virginia. We hypothesize that family experiences impact decision-making and access to care for children with appendicitis, are associated with increased rates of perforation (a proxy measure for delays in care), and that family experiences differ across populations. To test this hypothesis, we will use a cross sectional, mixed-methods approach to define parental and child experiences when accessing care for appendicitis. The goal of this report is to describe our study design, with the intent to disseminate this information and solicit feedback to enhance study performance. The objective of our study is to define family experiences in children with appendicitis through the following aims:

**Aim 1**: To describe family experiences and decision-making when accessing care for children with appendicitis. We will first record demographic data, clinical care, and clinical outcomes for consecutive children with appendicitis at two study sites. Second, we will collect quantitative data from parents/guardians of these children using surveys to assess response to childhood symptoms, adverse childhood experiences (ACEs), SDOH, and health literacy. Third, we will use nested purposive sampling to select a subset of families for qualitative interviews to explore in-depth key family experiences. We will collect parallel data from older children (ages 8–17 years). Finally, we will integrate all quantitative and qualitative data to inform a novel conceptual model of access to surgical care.

**Aim 2**: To evaluate if family experiences impact perforation rate in children with appendicitis. Our primary aim will test if parental responses to children's symptoms are associated with the rate of perforation (as a proxy measure for delays in care) using multivariate linear regression. Secondary aims will compare other parental experiences such a history of ACEs, health literacy, and SDOH on the rate of perforation.

**Aim 3**: To evaluate if family experiences differ across populations, including race, ethnicity, SES, and geographic location. Our primary aim will test if parental responses to children's symptoms differ across population groups. Secondary aims will explore if other parental experiences, such as a history of ACEs, health literacy, and SDOH, differ across populations and impact clinical outcomes.

## Materials and methods

### Theoretical framework and study design

To examine the root causes of surgical disparities in children, we will use the theoretical framework of Torain et al., which outlines the drivers of surgical disparities across multiple levels, including family/patient, provider, systemic, clinical care, and postoperative levels [15]. In line with ethical standards for research with vulnerable populations, we will use a strength-based framework to probe for protective factors associated with family resilience (e.g., social support, community cohesion) as well as identification of risk factors for delays in care and poor clinical outcomes [16].

We will use a prospective, convergent, mixed-methods, cross-sectional design to examine family experiences when accessing care for children with appendicitis (**specific aim 1**), to confirm if family experiences impact perforation rate (as a proxy measure of delays in care) (**specific aim 2**), and to compare family experiences across populations (**specific aim 3**) (Fig 1) [17]. Our mixed-methods design will attribute weight to qualitative data over quantitative data (QUAL + quant). Quantitative data will be based on data collected from medical records as well as standardized surveys for adults and children which examine family experiences which

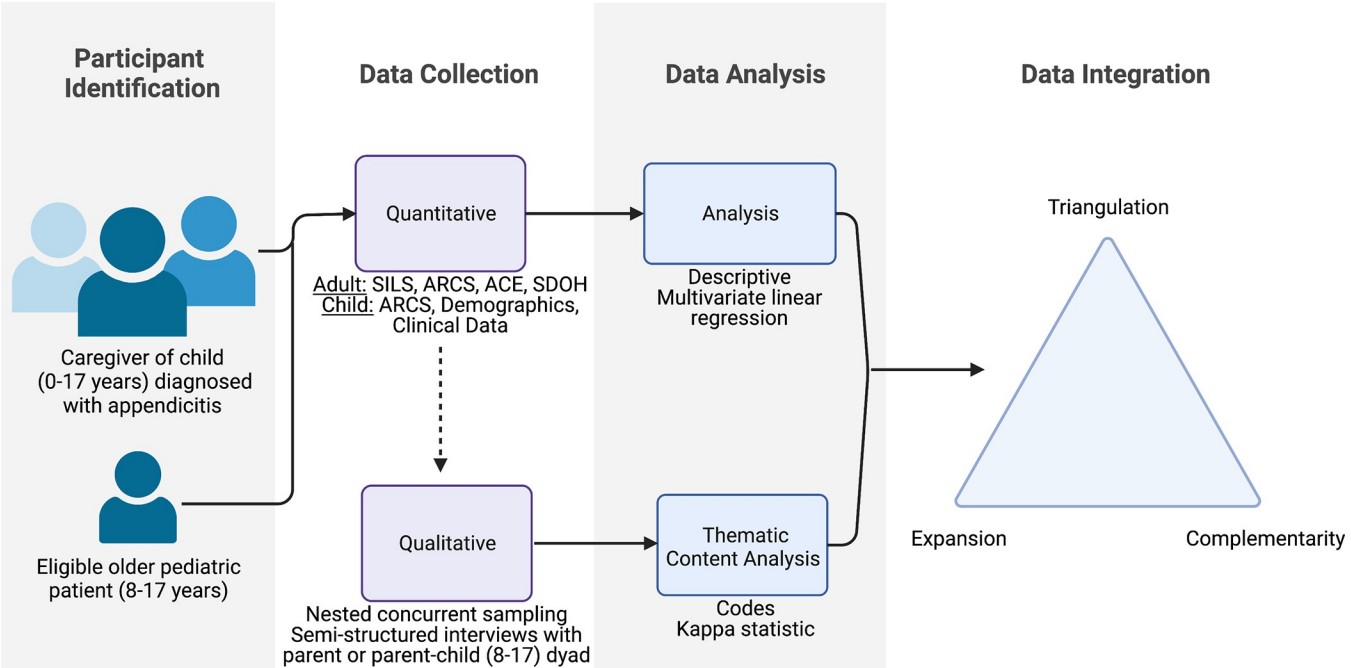

**Fig 1. Mixed methods study design to examine family experiences for children with appendicitis.** Participants will include the parent of a child with acute appendicitis and pediatric patients (8–17 years). All eligible adult participants will complete the 1) Adult Responses to Children's Symptoms (ARCS), 2) the Accountable Health Communities (AHC) Health-Related Social Needs Screening Tool which assesses social determinants of health (SDOH), 3) Adverse Childhood Experiences (ACE) of parent and child, and 4) Single Item Literacy Screener (SILS). Pediatric patients will be administered the child-report form of the Adult Responses to Children's Symptoms (ARCS) survey and may participate in a dyad interview. Demographic and clinical data will be collected from the child's electronic medical record and self-reported by the parent/caregiver. Using purposive, nested sampling, a subset of subjects will be selected to participate in semi-structured interviews (SSIs). Data will be analyzed using multivariate linear regression and integration will use triangulation, complementarity, and expansion to identify dominant themes of family experiences. Weight will be primarily attributed to qualitative data over quantitative data.

may impact access to care. Qualitative data will be collected from a subset of adult participants and children selected by nested concurrent, purposive sampling for semi-structured interviews to expand on quantitative findings (see nested sampling strategy).

We will integrate quantitative and qualitative data using a merging approach, in which the quantitative and qualitative datasets are brought together for triangulation and merged analysis. Integrated data will be used to inform a novel conceptual model of family experiences when accessing surgical care.

This study will be conducted according to Equator network guidelines, including Strengthening the Reporting of Observational Studies in Epidemiology (STROBE) [18] and the Consolidated Criteria for Reporting Qualitative Research (COREQ) [19].

## Participants

Enrollment will be offered to parents or legal guardians of consecutive children (ages 0–17 years) with appendicitis (confirmed at appendectomy and/or by radiographic imaging) at both study sites over the 15-month enrollment period. The enrollment period started on August 7, 2023, and is expected to end on November 7, 2024. We will collect additional data from older children (ages 8–17 years) so we can capture perspectives from children who are old enough to participate in research.

Eligible families (Table 1) will be identified by clinical staff, with families approached by study staff for enrollment. Assent and informed consent prior to enrollment will be obtained

**Table 1. Participant eligibility criteria for enrollment.**

| **Inclusion Criteria (Adult and Child)** |
| --- |
| Primary caregiver (parent, grandparent, legal guardian) of child (age 0–17 years) with appendicitis |
| Patient with confirmed appendicitis by appendectomy or by radiographic imaging |
| Patient presents to either study site for definitive management of appendicitis |
| English or Spanish Speaking |
| Child eligible for participation in survey & interview if 8–17 years old with appendicitis |
| **Exclusion Criteria** |
| Data collection not completed within 4 weeks after discharge for appendicitis |

in person or remotely. Following enrollment, all subjects will be assigned an identifier number to protect their confidentiality. Each participating family will designate one parent to participate in the study. Data collection will conclude within 4 weeks of hospital encounter to optimize recall. In return for their time, each adult participant will receive $20 after survey completion, and participants undergoing interviews will receive an additional $20.

## Ethics approval and consent to participate

If assent is given, parents or legally authorized representatives will provide verbal or written consent, depending on study site, to study team prior to enrollment. All methods will be performed in accordance with the relevant guidelines and regulations in observance of the Declaration of Helsinki. Study protocol was approved by the Duke University School of Medicine Research Ethics Board (Pro00106075) and Carilion Medical Center Research Ethics Board (IRB-22-1635). A summary of study results may be shared with participants who opt-in. Findings, including any protocol amendments, will be published in peer-reviewed journals and presented at scientific conferences.

## Demographic and clinical data

For all families, we will collect patient data from the electronic medical record (EMR), including patient demographic information using standardized definitions (sex, gender, race, ethnicity, SES, rurality), insurance status, duration of symptoms prior to presentation, diagnosis of perforated vs non-perforated appendicitis (confirmed by pathology and/or radiographic imaging), and dosage of pain (opioid, non-opioid) and other medications during emergency room stay [20]. We will also collect other variables to account for delays in care, including time and settings of healthcare access points prior to definitive presentation and time until surgery. For adult participants, we will collect additional self-reported demographic data (parental income, parental education level, marital status). For definitions of demographic variables, we will use self-reported race and ethnicity according to decennial census classifications from the EMR using definitions outlined by the Office of Management and Budget and the Office of Minority Health [21]. Rural versus urban categorization will be determined by the patient's primary address, with an RUCA code ≥4 identified as rural [22, 23].

Categorization of SES will be assigned based on parental self-report, using a combination of metrics including education attainment and family annual income. SES categories will be determined in a method similar as described by Sheffer et al., given its composite index based on granular family-level data, such as, income categories and education attaintment [24]. Education level will be classified into 5 categories with 1 as lowest and 5 as highest (< 12 years, high school graduate or equivalent, 13–15 years, college graduate, > 16 years). Income level will be divided into 7 categories with 1 as lowest and 7 as highest ($0–$9,999, $10,000–$24,999,

$25,000–$49,000, $50,000–$74,999, $75,000–$99,999, $100,000–$149,999, or $\geq$\$150,000). A combined score will result in a discrete SES score (range = 2–12). The scale will be divided into 3 SES categories (2–4, 5–8, & 9–12). If preliminary and interim analyses demonstrate that this division of SES results in bias split of the data, we will consider dividing income of participants enrolled in the study into lower and higher groups to balance SES status. As recommended by Braveman et al. [25], we will also collect other variables known to influence SES, such as, parental marital status [26], geographic indices, and subjective perceptions of SES through qualitative interviews [27]. These factors will be systematically considered in analysis, and qualitative findings may highlight unmeasured factors that affect conclusions.

## Survey instruments and interview guide

For all enrolled adult participants, we will administer five validated surveys that assess variables which have been associated with delays in care in other childhood health conditions, including asthma, obesity, and cancer [4]. These surveys include: 1) parental and child report forms of the Adverse Childhood Experiences (ACE) questionnaire, which contains 10 items to detail adverse childhood experiences [28]; 2) the Accountable Health Communities (AHC) Health-Related Social Needs Screening Tool, which is a widely used SDOH screening tool compiled by the Centers of Medicare & Medicaid Services which collects information across 13 domains, including housing instability, food insecurity, transportation difficulties, utility assistance needs, interpersonal safety, financial strain, employment, education, family and community support, physical activity, substance use, mental health, and disabilities [29, 30]; 3) the Single Item Literacy Screener (SILS), which rapidly assesses basic health literacy [31], and 4) the adult and child report forms of the Adult Responses to Children's Symptoms (ARCS) survey [32]. For older child participants (ages 8–17), we will administer the child report form of the Adult Responses to Children's Symptoms (ARCS). The ARCS is the most widely used instrument for assessing parental behavioral responses to a child's symptoms across four domains: protect, minimize, monitor, and distract. This survey has been validated in several conditions in children [33]. All surveys will be administered electronically via REDCap or on paper. All survey instruments will be available in English and Spanish, which are the dominant languages in the study settings.

We created a semi-structured interview tool for qualitative data collection in a format similar to Webster et al. [34], focusing on family experiences during access to care (S1 File). Our interview will probe for experiences about disease presentation, sources of health information, choice of hospitals, experiences accessing care, and barriers and facilitators to care. The interview guide will be pilot tested for face and content validity in 4–6 families. Separate questions for child participants will probe for areas of child-caregiver communication, interpretation of pain, and expression of symptoms. We will conduct 30–60-minute semi-structured interviews with the caregiver or caregiver-child dyad. Interviewers will be trained by the principal investigator at each study site.

## Nested sampling strategy

Following enrollment, we will use nested concurrent purposive sampling to select a subset of participants for in-depth qualitative interviews, with the goal to use these interviews to expand on findings from quantitative data. We will use a multilayered approach to ensure representative sampling. For example, we will first purposely sample towards perforation status, including at least 20 participants in perforated and non-perforated cohorts. Within each cohort (perforated/non-perforated), we will sample enough participants from each group (race, ethnicity, socioeconomic status, and rurality) to allow for adequate representation of our study

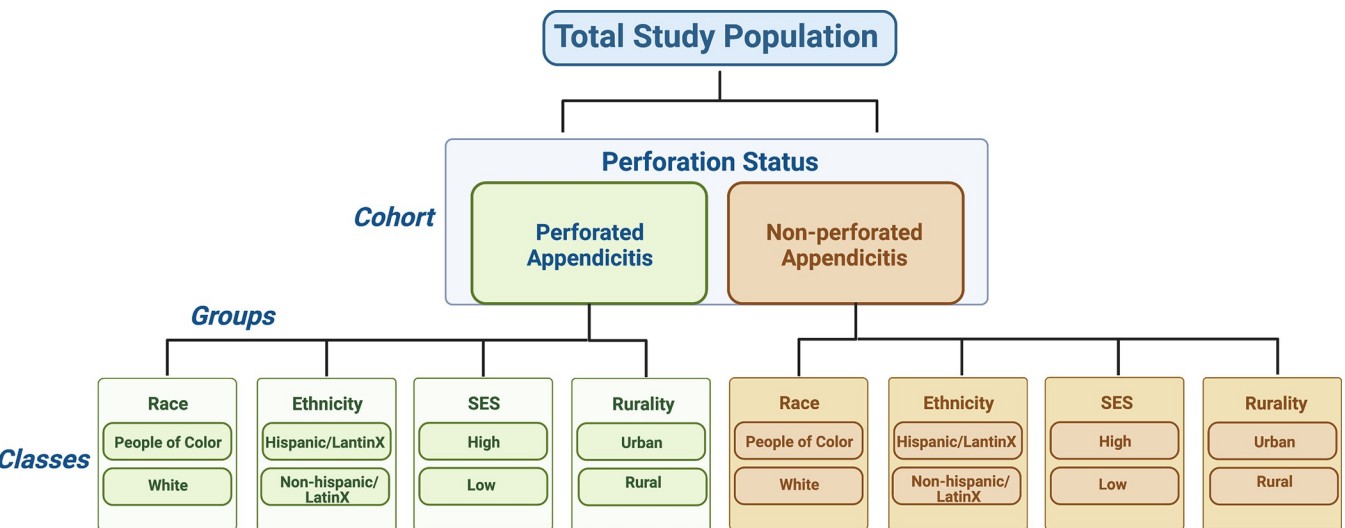

**Fig 2. Sampling strategy for qualitative interviews using nested, purposive sampling.** To compare family experiences across populations, we will interview at least 20 subjects within perforated and non-perforated cohorts (40 participants total). Based on demographics of the first interviewed participants, we will modify further sampling to ensure adequate representation for each group. If additional subjects are needed to attain data saturation for each group, we will conduct further sampling accordingly. SES: Socioeconomic status.

population (Fig 2). Note that as participants may have identities across multiple subgroups (i.e., by race, rurality, SES, etc.), we estimate that it will require ~40 total interviews to achieve representation across cohorts. If final analysis suggests a need for additional interviews to achieve data saturation, we will consider adding additional interviews. Depending on time and resources, we may expand the number of classes (e.g., low SES, high SES) within each group; however, we will start with a minimum of two classes within each group in line with existing literature that has illustrated disparities in care across these groups in other childhood health conditions. All interviews will be conducted in person or via video call, then recorded and transcribed.

## Data storage and management

To protect confidentiality, each adult and child participant will be assigned an alphanumeric study identification code. Only de-identified data, including interview transcripts, will be entered into a secure, joint REDCap (Research Electronic Data Capture) database with separate data access groups for each site [35]. Data definitions in REDCap will help ensure both sites uniformly abstract data from EMR. REDCap export and data analysis will take place on a secure analytic platform accessible only to the analysis team (SPARC, Storage and Programs Accelerating Research Collaborations).

## Data integration

We will integrate all qualitative and quantitative data using an organizing matrix to define dominant constructs impacting family experiences during access to surgical care. Weight will be primarily attributed to qualitative findings during data integration (QUAL + quant). We will employ three strategies to integrate qualitative and quantitative data. First, we will use triangulation as described by Farmer et al. [36] to evaluate convergence or divergence of themes from qualitative and quantitative sources. Second, we will compare results for complementarity of survey findings with qualitative interviews to add depth of understanding [37]. Third, we

will assess for expansion to evaluate how qualitative data may expand or explain any unanticipated findings in the quantitative results [38].

## Data analysis

**Specific aim 1.** Our hypothesis is that family experiences impact access to care for children with appendicitis. To describe family experiences when accessing care for children, we will collect clinical, quantitative surveys, and semi-structured interviews as described above. Quantitative and qualitative data will be integrated to inform a novel conceptual model of access to care and summarize how challenges are experienced differently across population groups.

Note that our qualitative data analysis is designed to define dominant themes which impact access to care. For example, we may identify parents of children with perforated appendicitis who report transportation as a major barrier. Of those parents of children with perforated appendicitis living in urban areas, they may identify reliability of public transportation as a concern.

For analysis of Likert-scale surveys, we will use the Mann-Whitney U non-parametric test. Cronbach's alpha will be calculated for each survey subscale, scale, and overall survey results to ensure internal consistency and reliability of interpretation of survey data [39]. To analyze qualitative interview data, we will perform thematic content analysis using coding, memo writing, and theme sorting. Thematic codes will be inductively and separately developed by two researchers based on the first 3–6 interviews to help develop a codebook for analysis. Codebook development will follow a three-cycle coding method [40]. The first cycle will use open, exploratory coding and verbatim coding to characterize participants' experiences. The second cycle will help develop axial codes by categorical, conceptual, and thematic organization of the array from first-cycle codes. Finally, focused coding will identify the most frequent and significant codes to inform salient data categories. NVivo will be used for data organization and the software and version reported.

To ensure durability of themes, two reviewers will independently assess transcripts from one interview and compare them with themes outlined by the researchers. An audit trial will be recorded with any changes in SSI questions and codebook. An additional study team member will code 20% of the interviews to ensure data quality in final coding. Intercoder reliability will be assessed and adjudicated to ensure a minimum kappa statistic of at least 0.8 (80% agreement). Integrated quantitative and qualitative data may inform subsequent participant checking or follow-up examination of outliers, significant results, or unexpected findings. A conceptual framework of patient and family-level factors impacting decision-making will be developed and summarized in an organizing matrix.

**Specific aim 2.** Our hypothesis is that key family experiences increase the rate of perforation in children with appendicitis. To confirm if family experiences impact the perforation rate (as a proxy measure of delays in care), our primary aim will test if parental responses to children's symptoms (ARCS survey) are associated with an increased rate of perforation. Secondary aims will compare other family experiences as measured in other quantitative surveys on the rate of perforation. We chose to assess responses to childhood symptoms as our primary aim given the importance of recognition of early symptoms for the prompt diagnosis and care of appendicitis.

We will create multivariate linear regression models to the effect and 95% confidence intervals (CIs) for associations between the rate of perforation (primary exploratory variable) and results of the parental ARCS survey (outcome variable). Covariates will include multiple confounding factors known to impact disparities in surgical care and delays in care, including

patient-level (age, sex, race/ethnicity, insurance status, etc.) and neighborhood-level (rurality, income, etc.) SDOH variables. An exhaustive model selection will be performed to determine the optimal model as defined by that with the lowest Bayesian Information Criterion (BIC). All analyses will be conducted using Excel and SPSS, SAS, or R and the software and version will be reported.

**Specific aim 3.** Our hypothesis is that family experiences for children with appendicitis differ across populations, including race, ethnicity, SES, and geographic location. Our primary aim will test if parental responses to children's symptoms differ across population groups using multivariate modeling using the analysis model as described in specific aim 2. Secondary aims will confirm if other parental experiences, such as a history of ACEs, health literacy, and SDOH, differ across populations.

## Power calculations

To evaluate if parental responses to children's symptoms impact the rate of perforation in children with appendicitis (specific aim 2) or differ across populations (specific aim 3), we conducted *a priori* power calculations to determine the minimum number of participants to achieve each aim. To determine if ARCS scores differ between parents of children with perforated appendicitis and non-perforated appendicitis (specific aim 2), our calculations determined that 172 adult subjects of the total enrolled (86 children with perforated appendicitis and 86 children with non-perforated appendicitis) would provide 90% power to detect differences in ARCS responses (protect, monitor, minimize, and distract) with $\alpha$ set at 0.05 (Cohen $d = 0.5$) using unmatched t test. To detect a difference in ARCS responses between and across all population groups (perforated/non-perforated, race, ethnicity, SES, rurality) (specific aim 3), our power calculations determined that 242 adult subjects (121 children with perforated appendicitis and 121 children with non-perforated appendicitis) would provide 90% power with the $\alpha$ set at 0.01 (Cohen $d = 0.5$) using unmatched t test [41]. Accounting for 5% attrition (due to missing data), we will enroll a maximum of 256 adults across both sites. Between the 2 study sites, clinical estimates report approximately 400 children receive an appendectomy each year.

For qualitative data, we expect 70% of thematic codes to be generated in the first 6 interviews and 92% after 12 interviews [37, 42]. We will perform an interim analysis at the halfway point to ensure there is sufficient data within and across groups.

## Discussion

To the best of our knowledge, our project is the first to use mixed methods to assess family experiences and decision-making while accessing surgical care for children. The innovative impact of this project is both practical and academic. From a practical standpoint, research on family experiences during access to surgical care can help clinicians and policymakers improve surgical equity for children. From an academic standpoint, the use of a mixed-methods design should markedly advance our understanding of the complex family experiences and decision-making processes when accessing surgical care for children and how these experiences differ across populations.

### Mixed methods design

The use of mixed methods offers several advantages over the use of quantitative data or qualitative approaches alone when examining family experiences during access to surgical care for children. The triangulation of findings on family experiences from multiple quantitative and qualitative sources enhances the convergent validity of any findings as the results (i.e., that the

findings are not simply a by-product of any single quantitative or qualitative method) [43]. By focusing on (quantitative) patterns of social determinants of health as well as (qualitative) explanations of how families access surgical care, we will capture the complexity inherent in family experiences when accessing care, and how the multiplicity of individual contexts influence[s] these experiences by different populations. The integration of quantitative and qualitative data will allow us to define how context and individual differences interact to influence the experiences of families of a child who requires surgical care, and how the processing of these experiences and impact on clinical outcomes may differ.

## Choice of primary outcome

Although many family experiences impact decisions to access surgical care, we chose to assess the parental response to childhood symptoms (ARCS survey) on the rate of perforation (specific aim 2) and how those responses differ across populations (specific aim 3) as our primary outcome given the importance of parental recognition of symptoms for the prompt diagnosis and treatment of appendicitis. As delays in care are directly related to an increased rate of perforation and adverse clinical outcomes, early recognition of symptoms of appendicitis is critical and represents a potential opportunity for the development of strategic interventions to reduce delays in care. We may find that parental responses as measured by the Adult Responses to Children's Symptoms (ARCS) survey show that parents who score higher on protective domain sub-scales (e.g., When you have abdominal pain, how often do your parent(s) pay more attention to you than usual?) present to the hospital more quickly after onset of symptoms and their child may have better outcomes [33, 44]. In qualitative interviews, parents may describe being more likely to seek care once the illness is perceived as severe; however, factors which influence perceived disease severity and response will likely differ according to the age of the child [14]. We anticipate qualitative data from the children about their interpretation of pain and communication to caregivers will help define the key factors that influence a parent's response to surgical illness.

## Nested sampling design

We chose to use a concurrent, purposive sampling strategy to guide qualitative data collection. Our goal is to use qualitative interviews to expand on findings from quantitative data through a structured data integration process [38]. This design allows us to interview a representative study population that can provide insight into these experiences. We will obtain representation across vulnerable demographic groups who have been shown in prior literature to experience disparities in other areas of pediatric care. In addition to purposive selection based on perforation status and demographics, we may also select additional participants for interviews based on significant quantitative results, outliers, or other exemplars of extreme cases to help explain findings of the quantitative data [43, 45]. Use of a combination of sampling strategies is often used in mixed-method studies of comparisons across populations and will allow us to examine important findings from the quantitative data in an in-depth, explanatory fashion [17].

## Novel conceptual framework

Overall, our goal is to understand key family experiences to help inform a novel conceptual model of family experiences and decision-making when accessing acute surgical care for children. An increased understanding of parental and child's values, experiences, and factors that influence decision-making processes during access to care can help develop evidence-based interventions to help guide families during access to care.

For example, we may find factors that inform improved shared decision-making between patients, family members, and health care providers to help with the provision of timely and high-quality surgical care for children, such as early discussion of options, risks, and benefits for imaging strategies. We recognize that shared decision-making is gaining increasing support among policy makers in many childhood conditions and is recommended by several pediatric expert bodies and regulatory organizations [46].

Several factors make health decision-making during access to care for children different from adults, particularly for acute surgical conditions such as appendicitis. Children's developmental context (e.g., biological, cognitive, psychosocial) as well as acute pain may limit their participation in health care decisions for surgical conditions [47]. Pediatric decision-making is also complicated by the inclusion of multiple drivers on health decisions, such as the child themselves, family members, extended social networks, social media, health care providers, etc. [48]. When faced with making decisions to seek surgical care on their child's behalf, parents must act as a surrogate, often deciding without full knowledge of "what would my child want?".

## Limitations

There are several challenges in this study. First, this analysis captures drivers at a single level of the family/child, although we will use family experiences to provide insights into drivers of health disparities across multiple levels, including the provider-level (e.g., implicit bias) or systemic-level (e.g., institutional policies, racism). Second, this study will be conducted using a single surgical condition at only two centers, which limits the generalizability of findings. Third, there is a risk of recall bias and post-hoc rationalization as families may not remember all the details of their experiences; however, we hope to minimize these risks by collecting all data within 4 weeks of hospital discharge. Fourth, there are always risks of observer biases conducting qualitative interviews. To mitigate these risks, we aim to have an interviewer matched to one of the participant's demographic groups (e.g., race, ethnicity, etc.). Fifth, the use of non-random purposive sampling may adversely affect the external validity (i.e., generalizability) of findings [17]. However, most quantitative research uses non-random sampling, and this approach enhances our ability to compare family experiences across populations. Sixth, we recognize that an individual's identity crosses multiple identifiers, and our design limits an intersectoral analysis. Finally, we recognize concerns of "patient blaming" when examining how social determinants of health influence family experiences, although our analysis will use the voices of families themselves in concert with SDOH measures to help define the barriers to surgical care.

## Conclusions

Our study should help identify key family experiences that impact decision-making when accessing care for appendicitis and explore if these experiences are associated with delays in care or differ across populations. Increased understanding of family experiences when seeking surgical care for children may inform novel strategies to mitigate surgical disparities in children, such as decision-making support tools or risk-tailored health information. Although beyond the capacity of our current study, our findings may lead to identification of implementation challenges for programs to improve care for children with surgical conditions. Potential barriers may include provision of high-quality health information, recognition of the patient's pain, low provider trust, and importance of power relations. We will incorporate these challenges into our conceptual framework to inform novel strategies to mitigate surgical disparities in children by identifying opportunities that are responsive to parents and children's needs.

## Supporting information

**S1 File. Adult/child dyad semi-structured interview guide.**
(PDF)

## Acknowledgments

The authors thank Stacy Murray, Cara Spivey, Kimberly Turnage, Allison McKell, Leslie LaConte, Michael Dieu, Hannah Zelinger, Emmy Duerr, and Thomas Shen for their support with regulatory and enrollment coordination at study sites.

## Author Contributions

**Conceptualization:** Bria Hall, Allison Tegge, Cesia Cotache Condor, Angelica Witcher, Elizabeth Creamer, Anna Tupetz, Emily R. Smith, Brian Meier, Henry E. Rice.

**Data curation:** Allison Tegge, Marie Rhoads, Mamata Reddy Tokala.

**Formal analysis:** Bria Hall, Allison Tegge, Henry E. Rice.

**Investigation:** Bria Hall, Cesia Cotache Condor, Marie Rhoads, Terri-Ann Wattsman, Brian Meier, Henry E. Rice.

**Methodology:** Bria Hall, Allison Tegge, Cesia Cotache Condor, Marie Rhoads, Angelica Witcher, Elizabeth Creamer, Anna Tupetz, Emily R. Smith, Brian Meier, Henry E. Rice.

**Project administration:** Bria Hall, Marie Rhoads, Terri-Ann Wattsman, Mamata Reddy Tokala, Brian Meier, Henry E. Rice.

**Resources:** Cesia Cotache Condor, Terri-Ann Wattsman, Brian Meier, Henry E. Rice.

**Software:** Allison Tegge, Mamata Reddy Tokala.

**Supervision:** Bria Hall, Marie Rhoads, Brian Meier, Henry E. Rice.

**Validation:** Bria Hall, Allison Tegge, Anna Tupetz, Brian Meier, Henry E. Rice.

**Visualization:** Bria Hall, Henry E. Rice.

**Writing – original draft:** Bria Hall, Henry E. Rice.

**Writing – review & editing:** Bria Hall, Allison Tegge, Cesia Cotache Condor, Marie Rhoads, Terri-Ann Wattsman, Angelica Witcher, Anna Tupetz, Emily R. Smith, Mamata Reddy Tokala, Brian Meier, Henry E. Rice.

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
