## [Decision Letter · Decision Letter 0]

1 Mar 2024

PONE-D-23-24866Understanding family-level decision-making when seeking access to acute surgical care for children: protocol for a cross-sectional mixed methods studyPLOS ONE

Dear Dr. Rice,

Thank you for submitting your manuscript to PLOS ONE. After careful consideration, we feel that it has merit but does not fully meet PLOS ONE’s publication criteria as it currently stands. Therefore, we invite you to submit a revised version of the manuscript that addresses the points raised during the review process. Please revise.

We look forward to receiving your revised manuscript.

Kind regards,

Academic Editor

PLOS ONE

Journal Requirements:

   "BH and MR received the standard graduate student funding from Virginia Tech Carilion School of Medicine to support subject compensation for this study (see funding letter for reference)."

Reviewers' comments:

Reviewer's Responses to Questions

**Comments to the Author**

1. Does the manuscript provide a valid rationale for the proposed study, with clearly identified and justified research questions?

Reviewer #1: Yes

Reviewer #2: Yes

2. Is the protocol technically sound and planned in a manner that will lead to a meaningful outcome and allow testing the stated hypotheses?

Reviewer #1: Yes

Reviewer #2: Yes

3. Is the methodology feasible and described in sufficient detail to allow the work to be replicable?

Reviewer #1: Yes

Reviewer #2: Yes

4. Have the authors described where all data underlying the findings will be made available when the study is complete?

Reviewer #1: Yes

Reviewer #2: Yes

5. Is the manuscript presented in an intelligible fashion and written in standard English?

Reviewer #1: Yes

Reviewer #2: Yes

6. Review Comments to the Author

You may also provide optional suggestions and comments to authors that they might find helpful in planning their study.

Reviewer #1: The is very useful and comprehensive study proposal with a well defined study design. The tools that will be used for data collection are well explained with the respect to the research questions.

However the authors should mention why data analysis plan for establishing the SES classification should be based on the work done by Sheffer et al (2012), reference 25

Reviewer #2: It is a very well-designed protocol and I agree that the mixed methods design will enhance the results of the study.

I don't know how the primary care is given in the 2 centers or if the patient's caregivers may consult a family practice or pediatrician before going to the emergency room.

Suppose the patient may seek for consulting early, but the primary-care physician delays the access to specialized care. In that case, it may be a confounding factor that is not evaluated in the protocol. So, maybe, besides the perforation rate it would be helpful to add another variable: the "time since symptoms began to first medical consulting".

Another factor that may influence perforation is an in-hospital delay of surgery. Suppose that, for any reason, the patient is not operated on promptly and the appendix may rupture. That’s not on the family experiences.

Other than that, it is a very good protocol, and I will be looking to find the results.

7. PLOS authors have the option to publish the peer review history of their article (what does this mean?). If published, this will include your full peer review and any attached files.

Reviewer #1: No

Reviewer #2: **Yes: **Eduardo Bracho-Blanchet

---

## [Author Response · Author response to Decision Letter 0]

15 Apr 2024

Journal Requirements: 

Manuscript format has been corrected to meet style requirements.

Thank you for this note. This has been amended. 

 "BH and MR received the standard graduate student funding from Virginia Tech Carilion School of Medicine to support subject compensation for this study (see funding letter for reference)."

The funders had no role in this study, and as advised, the amended Role of Funder statement is now included in our cover letter.

Corrected. Ethics statement moved to the methods section of manuscript. 

Reference list has been revised for completeness and proper formatting. Changes noted below:

References added: 

Reference for race and ethnicity definitions: US Department of Health and Human Services O of MH. Data collection standards for race, ethnicity, sex, primary language, and disability status. [cited 3 May 2022]. Available: https://minorityhealth.hhs.gov/data-collection-standards-race-ethnicity-sex-primary-language-and-disability-status

Reference for framework of SES data collection and analysis: Braveman PA, Cubbin C, Egerter S, Chideya S, Marchi KS, Metzler M, et al. Socioeconomic status in health research: one size does not fit all. JAMA. 2005;294: 2879–2888. doi:10.1001/jama.294.22.2879

Reference on development and domain description of SDOH screening tool: Billioux A, Verlander K, Anthony S, Alley D. Standardized screening for health-related social needs in clinical settings: The Accountable Health Communities Screening Tool. NAM Perspectives. 2017. doi:10.31478/201705b

REDCap citation: Harris PA, Taylor R, Thielke R, Payne J, Gonzalez N, Conde JG. Research electronic data capture (REDCap)—A metadata-driven methodology and workflow process for providing translational research informatics support. Journal of Biomedical Informatics. 2009;42: 377–381. doi:10.1016/j.jbi.2008.08.010

Reference removed: 

Erroneous remnant of a SDOH assessment tool that was considered but not incorporated into the protocol: Gottlieb L, Hessler D, Long D, Amaya A, Adler N. A randomized trial on screening for social determinants of health: the iScreen study. Pediatrics. 2014 Dec;134(6):e1611-1618.

Reviewers' comments:

Reviewer's Responses to Questions 

Comments to the Author

1. Does the manuscript provide a valid rationale for the proposed study, with clearly identified and justified research questions?

Reviewer #1: Yes

Reviewer #2: Yes

2. Is the protocol technically sound and planned in a manner that will lead to a meaningful outcome and allow testing the stated hypotheses?

Reviewer #1: Yes

Reviewer #2: Yes

3. Is the methodology feasible and described in sufficient detail to allow the work to be replicable?

Reviewer #1: Yes

Reviewer #2: Yes

4. Have the authors described where all data underlying the findings will be made available when the study is complete?

Reviewer #1: Yes

Reviewer #2: Yes

5. Is the manuscript presented in an intelligible fashion and written in standard English?

Reviewer #1: Yes

Reviewer #2: Yes

6. Review Comments to the Author

You may also provide optional suggestions and comments to authors that they might find helpful in planning their study.

Reviewer #1: The is very useful and comprehensive study proposal with a well defined study design. The tools that will be used for data collection are well explained with the respect to the research questions.

However the authors should mention why data analysis plan for establishing the SES classification should be based on the work done by Sheffer et al (2012), reference 25

Response: Great point. We have expanded on why we chose this SES classification for this population in the “demographic and clinical data section”. As illustrated in the manuscript text, we chose the composite index of SES using the method described by Shaffer et al (2012) due to its granular, family-level factors of income categories based on US Census Bureau median household income and discrete levels of educational attainment. Income, education level, and occupation are family-level measures commonly used in SES indices in the literature; however, Shaffer's approach incorporate the practical benefit of discrete categories to protect patient privacy (as opposed to direct income reporting) and allow for reconstruction of SES indices if interim analysis illustrates that the division resulted in a bias split of the data. Broader, population level measures of SES were considered, such as the Area Deprivation Index (ADI); however, as our study is not a large-scale epidemiological study, we chose to focus on SES status driven by family- level data. Lastly, Braveman et al (2005) highlighted the importance of analyzing individual socioeconomic factors in analysis when also using composite SES scores, as such, we are collecting other variables known to impact SES and will systematically account for this during analysis. Qualitative interviews will expand on subjective definitions of SES and may identify key unmeasured factors.

Reviewer #2: It is a very well-designed protocol and I agree that the mixed methods design will enhance the results of the study.

I don't know how the primary care is given in the 2 centers or if the patient's caregivers may consult a family practice or pediatrician before going to the emergency room.

Suppose the patient may seek for consulting early, but the primary-care physician delays the access to specialized care. In that case, it may be a confounding factor that is not evaluated in the protocol. So, maybe, besides the perforation rate it would be helpful to add another variable: the "time since symptoms began to first medical consulting".

Another factor that may influence perforation is an in-hospital delay of surgery. Suppose that, for any reason, the patient is not operated on promptly and the appendix may rupture. That’s not on the family experiences.

Other than that, it is a very good protocol, and I will be looking to find the results.

Response: We agree with this suggestion, and it will be important to adjust for these factors in the analysis. We accounted for this in the protocol, although not previously articulated in the manuscript, and are collecting several time-related variables, including family-reported variables (i.e., time onset of symptom, time to initial healthcare access point) and EMR-reported variables (i.e., ED arrival time, time of surgery, discharge time). These variables will enable our analysis team to adjust for other factors that drive delays in care. We are also collecting the contexts of “initial healthcare access point(s)” before ED presentation (pediatrician, outside emergency room, urgent care). We have further detailed the collection of these variables in our methods section, lines 233-235.

7. PLOS authors have the option to publish the peer review history of their article (what does this mean?). If published, this will include your full peer review and any attached files.

Do you want your identity to be public for this peer review? For information about this choice, including consent withdrawal, please see our Privacy Policy.

Reviewer #1: No

Reviewer #2: Yes: Eduardo Bracho-Blanchet

---

## [Decision Letter · Decision Letter 1]

8 May 2024

Understanding family-level decision-making when seeking access to acute surgical care for children: Protocol for a cross-sectional mixed methods study

PONE-D-23-24866R1

Dear Dr. Rice,

We’re pleased to inform you that your manuscript has been judged scientifically suitable for publication and will be formally accepted for publication once it meets all outstanding technical requirements.

Kind regards,

Academic Editor

PLOS ONE

Additional Editor Comments (optional):

Reviewers' comments:

Reviewer's Responses to Questions

**Comments to the Author**

1. Does the manuscript provide a valid rationale for the proposed study, with clearly identified and justified research questions?

Reviewer #1: Yes

Reviewer #2: Yes

2. Is the protocol technically sound and planned in a manner that will lead to a meaningful outcome and allow testing the stated hypotheses?

Reviewer #1: Yes

Reviewer #2: Yes

3. Is the methodology feasible and described in sufficient detail to allow the work to be replicable?

Reviewer #1: Yes

Reviewer #2: Yes

4. Have the authors described where all data underlying the findings will be made available when the study is complete?

Reviewer #1: Yes

Reviewer #2: Yes

5. Is the manuscript presented in an intelligible fashion and written in standard English?

Reviewer #1: Yes

Reviewer #2: Yes

6. Review Comments to the Author

You may also provide optional suggestions and comments to authors that they might find helpful in planning their study.

Reviewer #1: The authors have responded to reviewers' comments with acceptable explanation. The explanation have been added in the text and are useful for the readers.

Reviewer #2: With the writen response from the authors, I have no concerns and the protocolo should be accepted in my opinion.

7. PLOS authors have the option to publish the peer review history of their article (what does this mean?). If published, this will include your full peer review and any attached files.

Reviewer #1: No

Reviewer #2: **Yes: **Eduardo Bracho-Blanchet

---

## [Editor Report · Acceptance letter]

13 Jun 2024

PONE-D-23-24866R1 

PLOS ONE

Dear Dr. Rice, 

I'm pleased to inform you that your manuscript has been deemed suitable for publication in PLOS ONE. Congratulations! Your manuscript is now being handed over to our production team.

Kind regards, 

on behalf of

Dr. Robert Jeenchen Chen 

Academic Editor

PLOS ONE